# Antimicrobial Usage and Resistance in Companion Animals: A Cross-Sectional Study in Three European Countries

**DOI:** 10.3390/antibiotics9020087

**Published:** 2020-02-16

**Authors:** Philip Joosten, Daniela Ceccarelli, Evelien Odent, Steven Sarrazin, Haitske Graveland, Liese Van Gompel, Antonio Battisti, Andrea Caprioli, Alessia Franco, Jaap A. Wagenaar, Dik Mevius, Jeroen Dewulf

**Affiliations:** 1Veterinary Epidemiology Unit, Department of Obstetrics, Reproduction and Herd Health, Faculty of Veterinary Medicine, Ghent University, Salisburylaan 133, 9820 Merelbeke, Belgium; evelienodent@gmail.com (E.O.); s.sarrazin@lammerant.be (S.S.); jeroen.dewulf@ugent.be (J.D.); 2Department of Bacteriology and Epidemiology, Wageningen Bioveterinary Research, Houtribweg 39, 8221 RA Lelystad, The Netherlands; daniela.ceccarelli@wur.nl (D.C.); J.Wagenaar@uu.nl (J.A.W.); D.J.Mevius@uu.nl (D.M.); 3Department of Infectious Diseases and Immunology, Faculty of Veterinary Medicine, Utrecht University, Yalelaan 1, 3584 CL Utrecht, The Netherlands; haitskegraveland@hotmail.com; 4Institute for Risk Assessment Sciences, Utrecht University, Yalelaan 2, 3584 CM Utrecht, The Netherlands; L.VanGompel@uu.nl; 5Istituto Zooprofilattico Sperimentale del Lazio e della Toscana “M. Aleandri”, Department of General Diagnostics, National Reference Laboratory for Antimicrobial Resistance, Via Appia Nuova, 1411, 00178 Rome, Italy; antonio.battisti@izslt.it (A.B.); alessia.franco@izslt.it (A.F.)

**Keywords:** antimicrobial use, antimicrobial resistance, companion animals, critically important antimicrobials, colistin resistance, one health

## Abstract

Companion animals have been described as potential reservoirs of antimicrobial resistance (AMR), however data remain scarce. Therefore, the objectives were to describe antimicrobial usage (AMU) in dogs and cats in three European countries (Belgium, Italy, and The Netherlands) and to investigate phenotypic AMR. A questionnaire and one fecal sample per animal (*n* = 303) were collected over one year and AMU was quantified using treatment incidence (TI). Phenotypic resistance profiles of 282 *Escherichia coli* isolates were determined. Nineteen percent of the animals received at least one antimicrobial treatment six months preceding sampling. On average, cats and dogs were treated with a standard daily dose of antimicrobials for 1.8 and 3.3 days over one year, respectively. The most frequently used antimicrobial was amoxicillin-clavulanate (27%). Broad-spectrum antimicrobials and critically important antimicrobials for human medicine represented 83% and 71% of the total number of treatments, respectively. Resistance of *E. coli* to at least one antimicrobial agent was found in 27% of the isolates. The most common resistance was to ampicillin (18%). Thirteen percent was identified as multidrug resistant isolates. No association between AMU and AMR was found in the investigated samples. The issue to address, regarding AMU in companion animal, lies within the quality of use, not the quantity. Especially from a One-Health perspective, companion animals might be a source of transmission of resistance genes and/or resistant bacteria to humans.

## 1. Introduction

Antimicrobial resistance (AMR) is a complex issue with many contributing factors [1]. The most important contributing factor is antimicrobial usage (AMU), which can promote selection of bacteria with acquired resistance in animals and humans [2,3,4,5,6,7]. Moreover, transmission of resistant bacteria or their resistance genes is possible between animals and humans via direct or indirect contact, through food/feed and the environment [8,9,10]. For a long time, focus was mostly on food producing animals. This attention was translated into many actions and a significant reduction in AMU in many developed countries. Remarkably, this has been achieved without jeopardizing animal health and welfare in the food-producing industry [11,12].

Recently, dogs and cats are increasingly considered to also be a reservoir and a relevant transmission pathway [8,9,13,14]. The transmission of resistance through dogs and cats (further referred to as companion animals) to humans is described in scientific literature [15,16,17], but the extent to which this occurs is still largely unknown [18,19,20,21]. However, in Europe there are more than 102 million and 86 million households that own cats and dogs, respectively [22]. As it is known that owners often have intensive contact with their companion animals, the European Medicines Agency (EMA) already addressed the lack of knowledge regarding risk factors and transmission routes for transfer of AMR between companion animals and humans, also pointing out the limited understanding of current AMU in companion animals within the European Union’s (EU’s) member states [23].

Some studies in companion animals already demonstrated a high use of broad-spectrum antimicrobials, critically important antimicrobials for human medicine and a prescription behavior that is often not in line with current European guidelines [24,25,26,27,28,29]. Nevertheless, these type of studies are rather rare, mostly cover only one country and use different methods, which restricts comparison between countries. Multi-country studies with a standardized methodology, can improve our understanding of AMU in companion animals, which is key to enable the transition towards a more responsible AMU [23].

Prudent use of antimicrobials to prevent AMR is not only important from a public health perspective, but also for animal health and welfare [30]. It is expected that the loss of efficacy of antimicrobials, as seen in human medicine, will also arise in veterinary medicine [23]. Already, studies reporting on the isolation of resistant pathogens from cats and dogs, are becoming more common [31,32,33,34,35]. 

Not only does AMU have a selective pressure for resistance in pathogenic bacteria but also in commensals that are part of the microbiota [36,37]. Among the animal commensal flora, *Escherichia coli* has been recognized for humans as a potential source of exposure to resistance [38,39], and is considered a good indicator for the effects of selective pressure of AMU [40]. This cross-sectional study aims to explore the contribution of companion animals in the multifactorial ecology of AMR. The objectives were to describe and quantify in a standardized manner AMU in companion animals in three different European countries and to investigate phenotypic AMR in commensal *E. coli* isolated from fecal samples.

## 2. Results

### 2.1. Antimicrobial Use

The questionnaires of 303 companion animals (151 dogs and 152 cats) provided information on AMU in Belgium (dogs = 49; cats = 48), Italy (dogs = 50; cats = 50), and the Netherlands (dogs = 52; cats = 54). Over a one-year period, 19% (*n* = 58) of the animals received at least one antimicrobial treatment. In the three countries, frequency of animals with at least one antimicrobial treatment equaled 16%, 20%, and 22% for Italy, the Netherlands and Belgium, respectively. In total, cats received an antimicrobial treatment less frequently (13%) than dogs (25%), resulting in an odds ratio (OR) for dogs of 2.2 (95% CI: 1.2–4.1) when looking at species as a risk factor for being treated with antimicrobials. However, in Belgium 25% of the cats received at least one treatment in one year compared to 18% of the dogs. None of the other analyzed factors were identified as a significant predictor for AMU, except for one (Appendix A). Animals that always stay inside were less likely to receive an antimicrobial treatment (OR = 0.4; 95% CI: 0.2–09).

Taking into account the total sampled population (i.e., treated and not treated), treatment incidence (TI), which resembles the percentage of a full year that the animal has been treated with a standard dose of antimicrobials, ranged from 0.0% to 16.7% in cats and from 0.0% to 13.4% in dogs. The mean treatment incidence (TI) was 0.9% for dogs and 0.5% for cats, meaning that, on average, dogs and cats included in this study were being treated with a standard dose of antimicrobials during 3.3 days (0.9%) and 1.8 days (0.5%) in a full year, respectively. However, in Belgium, cats were more frequently exposed than dogs, with an average TI of 0.9 compared to 0.6, respectively. These results were in contrast to cats and dogs in the Netherlands and Italy, as the average TI in dogs was two times higher than in cats. Dogs in the Netherlands had the highest average TI (1.1), compared to Italian and Belgian dogs (0.8 and 0.6, respectively). A detailed overview on TI can be found in Table 1.

Of the animals that received antimicrobials, most of them received a single treatment (67%). For both species, amoxicillin-clavulanate was the most used active compound (cats 28%; dogs 27%). For cats, the second most used active compound was cefovecin (21%, *n* = 6), although this was not the case in every country as cefovecin usage for cats varied from 50% (*n* = 2) in Italy to 16% (*n* = 4) in Belgium and 0% (*n* = 0) in the Netherlands. In dogs, cefalexin and spiramycin-metronidazole (11% both) were the most chosen products, after amoxicillin-clavulanate. Figure 1 gives an overview on the frequency of usage for each active substance. The overall quantity of administered antimicrobials over a one-year period in this sampled population was 208,756.04 mg for dogs and 13,188.63 mg for cats.

Average treatment duration equaled 19 days for ofloxacin (one prescription), 19 days for metronidazole (2 prescriptions), and 14 days for cefovecin (7 prescriptions). The average treatment duration for oral administered antimicrobials equaled 10 days (avg ± sd = 10 ± 7 days; min–max = 1–42 days; 55 prescriptions), compared to 9 days for topical antimicrobials (avg ± sd = 9 ± 11 days; min–max = 1–42 days; 11 prescriptions) and 6 days for parenteral antimicrobials (avg ± sd = 6 ± 6 days; min–max = 1–14 days; 18 prescriptions).

### 2.2. Antimicrobial Resistance

*E. coli* was successfully isolated from 282 (93%) of the fecal samples, meaning isolation was unsuccessful for 21 samples. As a result, MIC values for a total of 282 *E. coli* isolates from Belgium (dogs = 44; cats = 44), Italy (dogs = 50; cats = 50), and the Netherlands (dogs = 51; cats = 43) were analyzed in this study. In total, 73% of the isolates were fully susceptible for the tested antimicrobials, meaning that resistance to at least one antimicrobial agent was found in 27% of the isolated *E. coli*. At country level, Italy had a higher number of resistant isolates (41%) than Belgium and the Netherlands (23% and 17%, respectively). As a result, when comparing to Italy, the odds to find a full susceptible isolate in Belgium and the Netherlands, were 2.7 (95% CI: 1.4–5.1) and 3.9 (95% CI: 2.0–7.8) times higher, respectively (a complete overview is shown in Appendix A). This shows significant differences in terms of full susceptibility between Italian isolates on one side and isolates from Belgium and the Netherlands on the other side. These general country differences were also mirrored by certain types of resistance. Resistance to ciprofloxazin (CIP) equaled 20% in Italy, compared to 5% and 1% in Belgium and the Netherlands, respectively. Looking at all isolates, the most common resistances in *E. coli* were to ampicillin (18%), sulfamethoxazole (15%), and tetracycline (14%). An overview for all tested antimicrobials, stratified per species and per country is shown in Figure 2 and Appendix A.

Two isolates showed resistance to colistin, a last resort antimicrobial in human medicine. For both isolates this was the only resistance detected. These isolates were both from the Netherlands, originating from a cat and a dog. The two animals did not have direct contact with each other. The cat did not receive any antimicrobial treatment in the year prior to sampling. The dog received four antimicrobial treatments in the previous year, of which one topical ear treatment with polymyxin B. No resistance to carbapenems (meropenem) or tigecycline was detected.

The most frequently found antimicrobial resistance patterns, stratified per country or species, are shown in Table 2, together with multidrug resistance (MDR). Thirteen percent was identified as MDR isolates, with resistance to three or more antimicrobial classes/subclasses (see definition for MDR in materials and methods). Most of the isolates were resistant to only one antimicrobial agent (34%), followed by resistance to two (18%) and six antimicrobials (16%) (Table 2). One isolate from Italy was resistant to 10 different antimicrobials, including resistance to cefotaxime, suggesting that the isolate is an extended-spectrum β–lactamase (ESBL) producer. Three other isolates, 2 Italian cats and 1 Belgian cat, showed a similar pattern with resistance to ampicillin, cefotaxime and ceftazidime. Prevalence of MDR isolates was significantly higher in cats (15.3%, *n* = 21) than in dogs (11%, *n* = 16) (OR = 1.8; 95% CI: 1.0–3.3). However, at the country level this difference was only significant in Italy (OR = 2.8; 95% CI: 1.1–8.3). The prevalence of MDR isolates at the country level was higher in Italian isolates (23%), compared to 11% and 5% in isolates from Belgium and the Netherlands, respectively. However, when looking at country as a risk factor to find a MDR isolate in companion animals, none of the countries seemed to have a significant higher risk (Appendix A).

Considering the population treated with antimicrobials in the year preceding sampling and a successful *E. coli* isolation (*n* = 55), 15 *E. coli* isolates (27%) showed resistance to at least one antimicrobial. Within this group of resistant isolates, 8 (57%) originated from animals that were treated with antimicrobials within 150 days before sampling. From one animal, the date of the last registered treatment was missing. Of the 40 non-resistant isolates within the group of treated animals, 50% were from animals that were treated more than 150 days before sampling. The OR to find a resistant *E. coli* isolate when having had an antimicrobial treatment less than 150 days before sampling equaled 1.3 (95% CI: 0.4–4.5). Considering the non-treated population one year before sampling and a successful *E. coli* isolation (*n* = 227), 64 *E. coli* isolates (28%) showed resistance to at least one antimicrobial. The OR of finding a resistant *E. coli* in the treated versus non-treated group was 1.0 (95% CI: 0.5–1. 9). Only considering the cat population, 39% from the treated population and 29% from the non-treated population, were resistant to at least one antimicrobial (OR= 1.9; 95% CI: 0.7–4.9). For dogs, prevalence of resistance to at least one antimicrobial amongst the treated and non-treated population equaled, 22% and 26%, respectively (OR = 0.7; 95% CI: 0.3–1.7).

## 3. Discussion

To our knowledge, this is the first multi-country study that reports on both AMU and AMR in companion animals, using standardized sampling, laboratory methods and AMU quantification protocols. Currently there is no binding European policy that requires countries to report their veterinary AMU for companion animals. Yet, this will become mandatory for all member states of the European Union by 2030 at the latest [41]. Concerning AMR monitoring, European legislation provides specific guidelines for food-producing animals; however, companion animals are not included in this monitoring [42].

Over one year, approximately one out of five animals in this study received at least one antimicrobial treatment. The observed average TI expressed per 100 animals-days at risk, is rather low (0.5 in cats, 0.9 in dogs), compared to intensive livestock production. Previous studies on European broiler and pig farms, using a similar quantification method, showed a median TI of 9.0 and 9.2 per 100 animal-days at risk, respectively [43,44]. However, this study shows an important use of critically important antimicrobials (71%) (CIAs; see Section 4.3 of the Materials and Methods) in cats and dogs. More specifically, 36% of the total number of treatments was represented by CIAs of high priority, the other 35% were CIAs of the highest priority. Comparable results are highlighted in other studies investigating AMU in companion animals [25,26,28,45,46,47,48]. A study in the UK, using data from 374 veterinary practices, reported similar proportions of CIAs usage, 60% in dogs and 81% in cats [49]. The most frequently used antimicrobial agent and CIA for dogs and cats was amoxicillin-clavulanate, which corresponds to previous research [26,27,28,50,51].

AMU showed significant variation between countries and species. Dogs in this study were more likely to be treated with an antimicrobial than cats. However, this was not the case in Belgium. At active substance level, there was a more frequent use of cefovecin, a CIA of highest priority, in cats (*n* = 6) compared to dogs (*n* = 1), as reported by Buckland and colleagues [49]. The most likely interpretation behind this difference could be the convenience of the product for usage in non-cooperative animals, as one parental administration of the product has a duration of action of up to 14 days [52].

At country level, the odds for a Belgian cat to be treated with an antimicrobial were on average 5.2 times higher than in Italy, and 3.2 times higher compared to a Dutch cat. These results are partly reflected in the cefovecin use in cats, with 16% (*n* = 15), 50% (*n* = 4), and 0% (*n* = 10) of the total number of treatments in cats being treatments with cefovecin in Belgium, Italy, and the Netherlands, respectively. An outcome that could be driven by country specific legislation on the restrictive use of CIAs. In 2014, such a legislation went into effect in The Netherlands, making antimicrobial sensitivity testing mandatory if the veterinarian wanted to use third generation cephalosporins, such as cefovecin [53]. To our knowledge, such a legislation for companion animals is absent in Belgium and Italy.

The relatively high usage of CIAs raises concerns about selection of resistance to these drugs in isolates from companion animals as potential source of resistance genes and/or resistant bacteria from companion animals to humans. Resistance prevalence data are similar to those observed in other studies [18,47,48]. In the current study, 27% of *E. coli* isolates from dogs and cats were resistant to at least one antimicrobial. Thirteen percent were classified as MDR and the most frequent observed resistance was to ampicillin. Compared to average resistance levels in food producing animals, resistance in cats and dogs seems to be rather low. Data on resistance in *E. coli* from European pigs has been collected in 2017 within the EU and summarized by the European Food Safety Authority (EFSA), allowing for a comparison of resistance levels found in this study. The EFSA report showed resistance levels between 50% and 70% for tetracycline, sulfamethoxazole, and trimethoprim [54], which is around three-times higher than the levels found for companion animals in this study.

Nevertheless, some resistance levels found in this study are worrisome, as two Dutch isolates were resistant to colistin [55]. To the best of our knowledge, colistin resistance in companion animals has only been described in China (2015) [9], Germany (2017) [56], Finland (2018) [57], Ecuador (2019), [58] and now also in the Netherlands. The Chinese study even reports a possible transmission from companion animals to humans [9]. The isolates in this study derived from one dog and one cat. Both animals were from the Netherlands, the only country that reported usage of this type of antimicrobial class (five events vs. zero events in both other countries). Only the dog received antimicrobial treatments in the previous year, of which one topical ear treatment with polymyxin B. Polymyxin B belongs to the same antimicrobial class as colistin and is often used in topical ear treatments for dogs and cats despite the fact that it belongs to the CIAs with highest priority. Although topical treatment with polymyxins will not directly select for the occurrence of colistin-resistant isolates in the gastrointestinal tract, the data reinforce that polymyxins should preferably not be used as a first-line treatment.

When comparing full susceptibility levels in companion animals in Belgium (77%), Italy (59%), and the Netherlands (83%), it seems that the odds of finding a fully susceptible *E. coli* isolate are lower in Italy. At species level, resistance in dogs from Belgium (20%), Italy (30%), and the Netherlands (24%), were comparable for the two neighboring countries, and only slightly higher in Italy. However, resistance in cats was twice as high in Italy (52%) compared to Belgium (25%), and more than five-times higher compared to the Netherlands (9%). A similar trend was observed for the prevalence of MDR in Italian cats (32%) compared to cats from Belgium (9%) or the Netherlands (2%). Interestingly, AMU in Italy, in this study, was not higher than in both other countries. More importantly, average TI in Italian cats (0.3) was the lowest compared to 0.5 in the Netherlands and 0.9 in Belgium. However, with the small sample size in this study, extrapolations of AMU and AMR to a national level should be done cautiously. In fact, general veterinary AMU levels in Italy do exceed those in Belgium and the Netherlands, when looking at the European Surveillance for Veterinary Antimicrobial Consumption (ESVAC) reports that cover all veterinary antimicrobials sold at national level in 30 European countries [59]. These higher usage levels in Italy, as reported by ESVAC, are present for both the reporting on sales of tablets (used in companion animals) and the reporting on other pharmaceutical forms (used mainly in food-producing animals, including horses) [59]. This raises the question if the higher prevalence of AMR and MDR observed in Italian cats, together with lower odds to find a fully susceptible Italian isolate compared to Belgium and the Netherlands, could be explained by this general higher AMU in veterinary medicine in Italy, and in such a case, why Italian cats seem to have such higher AMR and MDR levels compared to dogs. The latter is not the case for the Netherlands, where dogs show higher levels of AMR and MDR, although not significant. Further research, focusing on differences at species and country level in AMU and AMR in companion animals will be essential to answer these questions. In addition, future scientific research should also look into the AMU and AMR levels of the owners, as different studies have already suggested associations regarding AMR between companion animals and their owners [18,60,61,62,63].

The fact that AMU levels in this study do not completely comply with the trends seen in those numbers of the ESVAC reports representing the use in companion animals (sales of tablets) [59], is most likely due to the difference in type of data collected and quantification method. Quantification of AMU was based on the mean recommended daily dose, derived from the summary of product characteristics (SPC), to standardize the daily dosage as no ESVAC values exist for companion animals. The variation between the recommended daily dosages on the SPCs from different products with the same active compound was small. Nevertheless, the use of the recommended daily dose instead of the used daily dose (i.e., the exact amount the animal was prescribed) is less accurate to describe the real amounts of antimicrobials used. The weight of the animals was not specified in the questionnaires and therefore a standard weight was used. The same weights have been used in a recent Dutch study [28] and are based on a survey using over 40,000 dogs and 13,000 cats to calculate the mean weight of both species [64]. The biggest limitation of the collected data lies in the small sample size analyzed in our study, which makes extrapolation of the results to a national level challenging. In addition, the group of treated animals was small. Therefore, comparisons of AMU and AMR levels between countries and species should be interpreted while keeping in mind the restrictions of this small sample size. Nevertheless, the results did show similar trends with previous studies [25,26,28,45,46,47,48] and enabled the identification of some significant differences, although some might be missed due to a lack of power. A recent Dutch study reported on the many different factors influencing the antimicrobial prescribing behavior of veterinarians [52]. As no inclusion criteria were present at the level of the veterinarian, it is expected that a certain (selection) bias is present within the study.

A clear link between higher AMU and higher AMR could not be found within this study (OR, 1.0; 95% CI [0.5–1.9]). It must be emphasized that the sample size of treated animals was relatively small (*n* = 58). Additionally, the timing from antimicrobial treatment until sampling was not standardized and both topical treatments and short treatments were taken into account. Subgrouping different treatments (i.e., oral vs. injection, topical vs. systemic, longer duration vs. short duration) and comparing these groups and their effects on antimicrobial resistance was not possible because of the small sample size. Despite these restrictions of the collected data, the results do give us new valuable insights into AMU and AMR in cats and dogs.

## 4. Materials and Methods 

Data and samples were collected between January 2015 and February 2016 in the framework of the “Ecology from Farm to Fork Of microbial drug Resistance and Transmission (EFFORT)” project (http://www.effort-against-amr.eu/), which investigated the epidemiology and ecology of AMR in food-producing and companion animals, the environment and humans to quantify AMR exposure pathways for humans.

### 4.1. Participants 

A cross-sectional study was conducted in three European countries: Belgium, Italy and the Netherlands. Veterinary practitioners were contacted to participate in the study. These veterinarians randomly selected animals and aimed for a sample size of 50 dogs and 50 cats in each participating country. Companion animals, one per owner, were selected based on the following inclusion criteria: healthy animal, no veterinary visit 3 months before sampling, not suffering from any enteric disorder at the time of sampling, a minimum age of one year and not living on a farm. There were no selection criteria in terms of previous antimicrobial use, but a history record of AMU had to be available. A written informed consent from the participating owners was obtained. 

### 4.2. Sampling, Data Sources, and Measurement

One fresh fecal sample per animal was collected from the ground or litter box (minimum 10 g of feces per sample). The owners filled in a questionnaire, developed within the EFFORT consortium, containing questions about contact with other pets, diet, gastrointestinal problems, living area, medical treatment in the past, etc. (Appendix A). Fecal samples were aseptically stored in containers at 4 °C and transported to the lab for processing within 24 h.

### 4.3. Quantification of AMU 

The available information from the questionnaires concerning AMU per animal included: the commercial antimicrobial name, the frequency of administration (per day) and the duration of treatment. This was done for all antimicrobials administered to the animal during the year preceding the questionnaire (Appendix A). Treatments could be curative, or set up from a metaphylactic or prophylactic point of view. Additional information on active compound and administration route was retrieved from the summary of product characteristics (SPC). For each antimicrobial compound, a color code was assigned in accordance with the classification of the World Health Organization (WHO) list of CIAs [65]: green are important antimicrobials, yellow are highly important, orange are critically important antimicrobials of high priority and red are critically important antimicrobials of the highest priority. In this manuscript, CIAs refers to antimicrobials belonging to the last two classes of that list.

AMU quantification was based on the methodology used by previous studies within the EFFORT project [43,44]. However, TI was calculated by using a simplified formula as necessary assumptions lead to identical factors in both nominator and denominator (Table 3). A detailed description of the different factors can be found in the Appendix A.

### 4.4. E. coli Isolation and Identification 

Isolation methods were focused at *E. coli* as sentinel microorganism. Only one *E. coli* isolate per sample was included in the study. All fecal samples were directly inoculated on MacConkey agar plates and after an overnight incubation at 37 °C, one randomly picked presumptive *E. coli* colony per sample was isolated, identified according to local standard methods and/or stored individually in buffered peptone water with 20% glycerol at −80 °C, pending analysis.

### 4.5. Antimicrobial Susceptibility Testing 

Minimum inhibitory concentrations (MIC) with broth microdilution were determined for a fixed panel of antimicrobials by commercially available microtitre plates (EUVSEC, Thermo Fisher Scientific UK Ltd, Loughborough, UK) according to ISO standard 20776-1-2006 [50]. EUCAST epidemiological cut-off values were used to differentiate between wild-type and non-wild-type susceptibility (henceforward referred to as resistant isolates). The epidemiological cut-off values (ECOFFs) to define wild-type *E. coli* are listed in the Appendix A [51]. *E. coli* ATCC 25955 was used as internal control strain. An isolate was defined as resistant or non-fully susceptible when it showed resistance to at least one antimicrobial. In case of resistance to three or more antimicrobial classes/subclasses, the isolate was defined as MDR. If resistance was present for 10 or more antimicrobials, the isolate was classified as XDR, based on the definitions from Feltrin and colleagues [66].

### 4.6. Data Analysis and Statistical Analysis

The results of the questionnaires were exported to commercial software Excel (Microsoft Corporation, Redmond, Washington, DC, USA). The database was cleaned using the following procedure: treatments dated more than one year before the study were removed from the database. Data was anonymized to ensure that results cannot be traced back to individual animals and owners. After data cleaning, descriptive statistics were performed using Excel. To look for factors influencing AMU, a univariate logistic regression was performed with AMU as a binary (use vs. no use) dependent variable. Several parts of the questionnaire were taken into account after transformation of the relevant questions into factors (Appendix A). When significant, OR and their corresponding 95% confidence intervals (CI) were estimated to describe the relationship with AMU (Table 2). Non-significant factors are not shown. An identical logistic regression was performed to explore the influence of having had an antimicrobial treatment (binary), on the presence of AMR in the *E. coli* isolated from the fecal sample. Within the group of the treated animals, an additional binary summary factor, named “last treatment less than 150 days before sampling”, was created to see if the period between the last reported treatment and the moment of sampling had any influence on the occurrence of resistance. OR and their corresponding 95% confidence intervals (CI) for this second logistic regression are shown in the results. All data analysis was performed in R version 3.4.0 software (https://cran.r-project.org).

## 5. Conclusions

In conclusion, with 81% of the animals not receiving any antimicrobial treatment over a year, AMU in companion animals seems rather low compared to AMU on pig and poultry farms [43,44]. Nevertheless, when an antimicrobial was used, it was often a CIA. For this reason, the issue to address, regarding AMU in companion animal, does not lie so much within the quantity but rather within the quality of antimicrobials used. Especially from a One-Health perspective, companion animals might be a source of transmission of resistance genes and/or resistant bacteria to humans. At country level, higher resistance in companion animals seems to follow trends of higher antimicrobial use (both food-producing and companion animals), as reported by ESVAC [59], suggesting that resistance levels in companion animals might be driven by other factors than only direct selective pressure by antimicrobial treatment in cats and dogs.

## Figures and Tables

**Figure 1 antibiotics-09-00087-f001:**
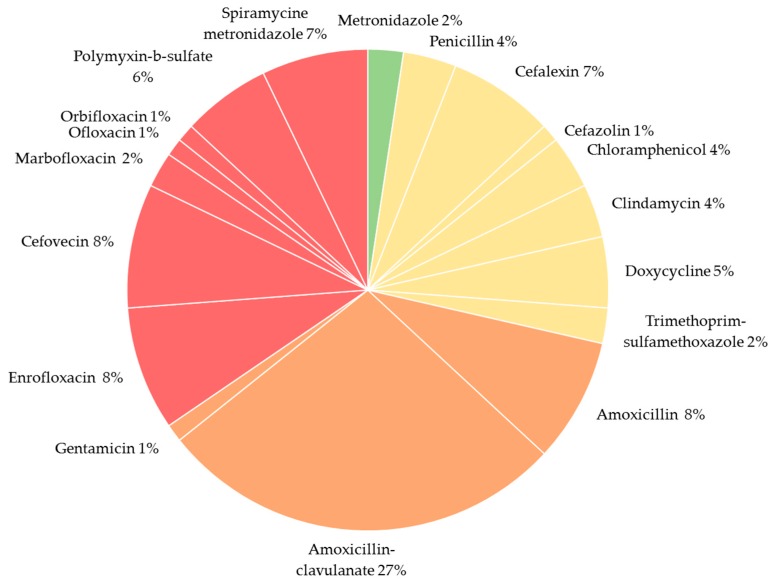
Percentage of antimicrobial treatments per active compound. Every piece of the pie chart is colored in the corresponding assigned color code based on the classification of the World Health Organization (WHO). Green color = important antimicrobial; yellow color = highly important antimicrobial; orange color = critically important antimicrobial of high priority; red color = critically important antimicrobial of highest priority; Results at species and country level are shown in Appendix A.

**Figure 2 antibiotics-09-00087-f002:**
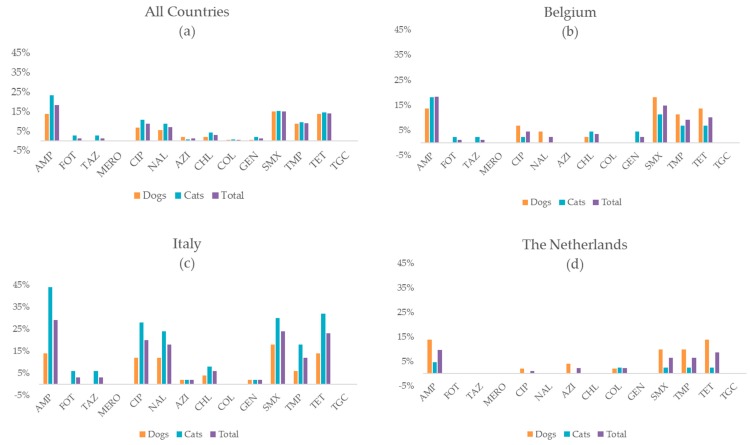
Antimicrobial resistance proportions (%) among *Escherichia coli* isolates from feces of dogs and cats in three European countries against a set of 14 antimicrobials. (**a**) Shown for all samples with a successful *E. coli* isolations from cats (*n* = 137) and dogs (*n* = 148); (**b**) *E. coli* from Belgian cats (*n* = 44) and dogs (*n* = 44); (**c**) *E. coli* from Italian cats (*n* = 50) and dogs (*n* = 50); (**d**) *E. coli* from Dutch cats (*n* = 43) and dogs (*n* = 51). AMP = ampicillin; FOT = cefotaxime; TAZ = ceftazidime; MERO = Meropenem; CIP = ciprofloxacin; NAL = nalidixic acid; AZI = azithromycin; CHL = chloramphenicol; COL = colistin; GEN = gentamicin; SMX = sulfamethoxazole; TMP = trimethoprim; TET = tetracycline; TGC = tigecycline. (Appendix A).

**Table 1 antibiotics-09-00087-t001:** Antimicrobial usage in companion animals expressed as the sum of treatment incidence (TI) at animal level.

Country	Log. Regression OR^1^ β—*p*-Value	Subgroup OR^1^ [95% CI]	Study Population Total (%Treated–%Non-Treated)	Log. Regression OR^2^ β—*p*-Value	OR^2^ [95% CI]	TI Avg (Min–Max)
**BE ^a^**		Cats	48 (25–75%)	1.7–0.01	5.2 [1.5–24.2] ^a^	0.9 (0.0–11.5)
		Dogs	49 (18–82%)	−0.4–0.4	0.6 [0.2–1.7] ^a^	0.6 (0.0–13.4)
	**−0.4–0.4**	**0.7** [0.2–1.8]	**97 (22–78%)**	**0.4–0.3**	**1.4 [0.7–3.0] ^a^**	**0.7 (0.0–13.4)**
**IT ^b^**		Cats	50 (6–94%)	0.5–0.5	1.5 [0.4–8.1] ^b^	0.3 (0.0–7.1)
		Dogs	50 (26–74%)	0.2–0.6	1.3 [0.5–3.0] ^b^	0.8 (0.0–11.0)
	**1.7–0.01**	**5.5** [1.6–25.3]	**100 (16–84%)**	**0.3–0.5**	**1.3 [0.6–2.7] ^b^**	**0.5 (0.0–11.0)**
**NL ^c^**		Cats	54 (9–91%)	1.2–0.04	3.2 [1.1–11.0] ^c^	0.5 (0.0–16.7)
		Dogs	52 (31–69%)	−0.7–0.2	0.5 [0.2–1.3] ^c^	1.1 (0.0–7.1)
	**1.5–0.008**	**4.4** [1.5–14.3]	**106 (20–80%)**	**0.1–0.7**	**1.1 [0.6–2.2] ^c^**	**0.8 (0.0–16.7)**
**Total**		Cats	152 (13–87%)			0.5 (0.0–16.7)
		Dogs	151 (25–75%)			0.9 (0.0–13.4)
	**0.8–0.009**	**2.2** [1.2–4.1]	**303 (19–81%)**			**0.7 (0.0–16.7)**

Avg = Average; [95% CI] = 95% Confidence Interval; Min = Minimum; Max = Maximum. BE = Belgium; IT = Italy; NL = the Netherlands; OR^1^ shows the odds ratio for species as a risk factor, at country level and overall; OR^2^ shows the odds ratio for country as a risk factor, at species level and overall. ^a,b,c^ comparison between two countries: ^a^ IT–BE, ^b^ IT–NL, ^c^ NL–BE.

**Table 2 antibiotics-09-00087-t002:** Phenotypes of resistance among *Escherichia coli* isolated from feces of dogs and cats in three European countries.

Nr of AM	Nr of Isolates (%)	Nr of Isolates from cats (%)	Nr of Isolates from Dogs (%)	Nr of Isolates in BE (%)	Nr of Isolates in IT (%)	Nr of Isolates in NL (%)	Most Frequent Pattern
**0**	205 (74%)	96 (70%)	109 (75%)	68 (77%)	59 (59%)	78 (83%)	-
**1**	26 (9%)	15 (11%)	11 (8%)	8 (9%)	11 (11%)	7 (7%)	AMP
**2**	14 (5%)	5 (4%)	9 (6%)	3 (3%)	7 (7%)	4 (4%)	SMX-TET; AMP-SMX
**3**	11 (4%)	5 (4%)	6 (4%)	1 (1%)	7 (7%)	3 (3%)	AMP-SMX-TMP
**4**	5 (2%)	3 (2%)	2 (1%)	2 (2%)	2 (2%)	1 (1%)	AMP-SMX-TMP-TET
**5**	6 (2%)	3 (2%)	3 (2%)	4 (5%)	2 (2%)	-	AMP-CHL-SMX-TMP-TET
**6**	12 (4%)	8 (6%)	4 (3%)	2 (2%)	9 (9%)	1 (1%)	AMP-CIP-NAL-SMX-TMP-TET
**7**	2 (<1%)	1 (<1%)	1 (<1%)	-	2 (2%)	-	AMP-CIP-NAL-AZI -SMX-TMP-TET; AMP-CIP-NAL-CHL-SMX-TMP-TET
**10**	1 (<1%)	1 (<1%)	-	-	1 (1%)	-	AMP-FOT-TAZ-CIP-NAL-AZI-GEN-SMX-TMP-TET
Total	282	137	145	88	100	94	

Nr = number; AM = antimicrobials; BE = Belgium; IT = Italy; NL = the Netherlands; AMP = ampicillin; AZI = azithromycin; FOT = cefotaxime; TAZ = ceftazidime; CIP = ciprofloxacin; CHL = chloramphenicol; GEN = gentamicin; NAL = nalidixic acid; SMX = sulfamethoxazole; TET = tetracycline; TMP = trimethoprim.

**Table 3 antibiotics-09-00087-t003:** Formulas to quantify antimicrobial usage using treatment incidence (TI).

Formula		Result
**TI_DDDca_^1^**	$\;=\frac{\mathrm{AAD}\;\left(\mathrm{mg}/\mathrm{kg}/\mathrm{day}\right)\;\times \;\mathrm{treatment}\;\mathrm{duration}\;\left(\mathrm{days}\right)\;\times \;\mathrm{standard}\;\mathrm{weight}\;\mathrm{per}\;\mathrm{animal}\;\left(\mathrm{kg}\right)}{\mathrm{DDDca}\;\left(\mathrm{mg}/\mathrm{kg}/\mathrm{day}\right)\;\times \;\mathrm{number}\;\mathrm{of}\;\mathrm{days}\;\mathrm{at}\;\mathrm{risk}\;\left(\mathrm{days}\right)\;\times \mathrm{standard}\;\mathrm{weight}\;\mathrm{per}\;\mathrm{animal}\;\left(\mathrm{kg}\right)}\;\times \;\mathrm{LA}\;\mathrm{factor}\;\times \;100\;\mathrm{AAR}$	Number of DDDca/100 days at risk/animal
**TI_DDDca_^2^**	$=\frac{\mathrm{treatment}\;\mathrm{duration}\;}{\mathrm{number}\;\mathrm{of}\;\mathrm{days}\;\mathrm{at}\;\mathrm{risk}\;}\times \;\mathrm{LA}\;\mathrm{factor}\;\times \;100\;\mathrm{AAR}$	Number of DDDca/100 days at risk/animal

ADD = Assumed Administered Dose; DDDca = Defined Daily Dose for companion animals; LA factor = long acting factor; AAR = animals at risk; ^1^ The original TI formula for an individual antimicrobial treatment; ^2^ The simplified TI formula with all identical factors in nominator and denominator removed.

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
