# Peer review of "Antimicrobial Usage and Resistance in Companion Animals: A Cross-Sectional Study in Three European Countries"

_antibiotics, 2020, doi:10.3390/antibiotics9020087_

Round 1
Reviewer 1 Report
The authors of this manuscript describe an international collaboration to determine the antimicrobial use in companion animals. They also go further to determine the antimicrobial resistance in a single species of bacteria isolated from faecal samples. In total 303 questionnaires and faecal samples were collected during the study. The collaboration was between three European countries: Belgium, Italy and The Netherlands. The sample size was fairly small but understandably so as there is no network in place to support this kind of study. Overall, the manuscript is written to a good publication standard with the exception of a few minor errors. The description of the Materials and Methods is long and could be condensed or moved to Supplementary if possible. The E. coli isolation and antimicrobial testing use standard methods which could be referenced to shorten. The subject of the manuscript and the data produced is novel and is of high scientific interest. During the study, two isolates with resistance to colistin were found which is significant. The authors do not overstate the findings of study and their conclusions are in-line with the data presented. I would recommend this manuscript for publication.
Changes:
Sentence starting line 26: Name the countries involved in the study in the Abstract. For example, “Therefore, objectives were to describe antimicrobial usage (AMU) in dogs and cats in three European countries (Belgium, Italy and The Netherlands) and to investigate phenotypic AMR.
Sentence starting line 45: Delete sentence “Bacteria have the ability to adapt quickly to their environment [1].” It doesn’t fit with the sentence before or after which complement each other.
Figure 1: insert space between Gentamicin and 1%. Also, I believe Trimethoprim-sulphonamide and 2% has double space.
Sentence starting 120: suggested change: “The longest average treatment duration was for oxfloxacin…..”
Sentence starting 127: Change to: “E. coli was successfully isolated from 282 (93%) of the faecal samples.”
Line 289: Remove markup capital T
Line 298: replace “gr” with “g”.
Table 3. Check formatting as the equation is cropped.
Author Response
Comments and Suggestions for Authors: Reviewer 1
The authors of this manuscript describe an international collaboration to determine the antimicrobial use in companion animals. They also go further to determine the antimicrobial resistance in a single species of bacteria isolated from faecal samples. In total 303 questionnaires and faecal samples were collected during the study. The collaboration was between three European countries: Belgium, Italy and The Netherlands. The sample size was fairly small but understandably so as there is no network in place to support this kind of study. Overall, the manuscript is written to a good publication standard with the exception of a few minor errors. The description of the Materials and Methods is long and could be condensed or moved to Supplementary if possible. The E. coli isolation and antimicrobial testing use standard methods which could be referenced to shorten. The subject of the manuscript and the data produced is novel and is of high scientific interest. During the study, two isolates with resistance to colistin were found which is significant. The authors do not overstate the findings of study and their conclusions are in-line with the data presented. I would recommend this manuscript for publication.
Thank you for this positive feedback and the comments made to improve this manuscript. We understand that the Materials and Methods section is rather long. However, we believe this is essential information for the readers, depending on their background. To shorten this part of the manuscript without losing this information, we moved some parts to the supplementary materials. We hope these adaptations are received as a good compromise. [Part G supplementary materials]
Changes:
Sentence starting line 26: Name the countries involved in the study in the Abstract. For example, “Therefore, objectives were to describe antimicrobial usage (AMU) in dogs and cats in three European countries (Belgium, Italy and The Netherlands) and to investigate phenotypic AMR.
Thank you for this suggestion, we adjusted the manuscript accordingly. [line 27]
Sentence starting line 45: Delete sentence “Bacteria have the ability to adapt quickly to their environment [1].” It doesn’t fit with the sentence before or after which complement each other.
Thank you for this suggestion. We agree with this comment and adjusted the manuscript accordingly.
The manuscript now reads: “Antimicrobial resistance (AMR) is a complex issue with many contributing factors [1]. The most important contributing factor is…” [line 47]
Figure 1: insert space between Gentamicin and 1%. Also, I believe Trimethoprim-sulphonamide and 2% has double space.
Thank you for noticing this error, we have adjusted figure 1 accordingly [see figure 1].
Sentence starting 120: suggested change: “The longest average treatment duration was for oxfloxacin…..”
Thank you for this suggestion, we adjusted the manuscript accordingly. [line 122-123]
Sentence starting 127: Change to: “E. coli was successfully isolated from 282 (93%) of the faecal samples.”
Thank you for this comment, we changed the sentence according to your suggestion with some minor adaptations to link this sentence to the rest of the paragraph. The revised manuscript now reads: “E. coli was successfully isolated from 282 (93%) of the faecal samples, meaning isolation was unsuccessful for 21 samples.. As a result, MIC values…” [line 130-131]
Line 289: Remove markup capital T
Thank you for noticing this error, we have removed the old track changes from the revised manuscript. [line 313]
Line 298: replace “gr” with “g”.
Thank you for noticing this error, we adjusted the manuscript according to the comment. [line 322]
Table 3. Check formatting as the equation is cropped.
Again, thank you for noticing this. The final part of the equation was indeed cropped. The necessary adaptions have been made, the full equation is now shown in Table 3. [line 343]
Reviewer 2 Report
Summary: This study is an excellent source of data regarding companion animal AMU and AMR in three European countries. Although the dataset is limited, it provides important context as part of AMR in a One Health paradigm. My comments are generally minor and are listed below.
Major comments:
Need to include some limitations of the study, such as the relatively few vets and specific veterinary settings potentially limiting the overall applicability of the data.
Take care when analyzing the data, given the relatively few animals that were actually treated with antimicrobials. This should be a caveat listed throughout, particularly when comparing data between countries or between dogs and cats, since these comparisons further stratify the data.
Sequencing data would have been ideal, with some comparisons of resistance mechanisms and phylogeny with E. coli from other sources, but this is not an absolute requirement to meet the study’s goals.
Specific comments:
Abstract: Should add mention that no association between AMU and AMR was found, while cautioning that several factors complicate this analysis. Still, it is an important finding given that AMU and AMR are measured in this study, and should be mentioned in the abstract.
L96-8: Should more clearly define in text and/or table that treatment incidence is the % of the year that a dog or cat is on average treated. Recommend rephrasing to say something such as:
Cats were treated from 0.0-16.6% of the year’s duration, and 0.0-13.4% of the year for dogs. On average, dogs were treated for 0.9% of the year (3.3 days), and cats 0.6% (1.8 days), with these numbers reflecting the treatment incidence (TI) in Table 1.
L100: This line belongs in the previous paragraph, where it was already stated that 25% of cats in Belgium were treated.
L109-110: This sentence isn’t clear, as it makes it sound as if cefalexin and spiramycin-metronidazole were the ‘most chosen products’ instead of amoxi-clav
Figure 1: Check each of the drug names; I think at least one is wrong (is it supposed to be trimethoprim-sulfamethoxazole?), with several having an extra ‘e’ at the end orbifloxacin, ofloxacin, clindamycin).
L122: Would not recommend saying ‘longer’, considering these differences are not statistically significant. Would simply state the average durations without comment.
L132: Don’t understand the point being made with this clause that begins ‘with significant differences…’. Recommend rephrasing.
L149: This definition of MDR as three drugs is different than in the methods, which lists three or more classes. Please clarify.
L196: add: ‘, respectively’ at end of this sentence to clarify to which animals the 9.0 and 9.2 refer
L204: Unclear what ‘Next to common characteristics’ refers to; recommend deleting
Discussion: Need to mention overall limitation of AMU data, with only 20 cats and 38 dogs treated overall. This means some variation in the data and additional limitations when further stratified by country.
L232: E. coli with mcr-1 was also found in a dog in Germany: https://www.ncbi.nlm.nih.gov/pubmed/28122910
Add discussion of AMU in people, since they may be more likely to share microbes than agricultural animals with companion animals (especially since animals on farms were not included in this study).
L262-5: I do not understand this sentence, since the ESVAC reports focus on food animals, correct? It is therefore not expected for that report to be relevant to this study focusing on companion animals.
L372: ‘levels of AMU in companion animals seem rather low’ is an ambiguous statement of limited value. I recommend stating that ‘x% of animals did not have antimicrobial usage over a one year period. However, when they were used, they were often CIA…’
Need to add the additional limitations of not including different types of veterinary care facilities (e.g. animal hospitals, etc) that may have dramatically different AMU (including with drugs administered IV, etc).
Author Response
Comments and Suggestions for Authors: Reviewer 2
Summary: This study is an excellent source of data regarding companion animal AMU and AMR in three European countries. Although the dataset is limited, it provides important context as part of AMR in a One Health paradigm. My comments are generally minor and are listed below.
Major comments:
Need to include some limitations of the study, such as the relatively few vets and specific veterinary settings potentially limiting the overall applicability of the data. Take care when analyzing the data, given the relatively few animals that were actually treated with antimicrobials. This should be a caveat listed throughout, particularly when comparing data between countries or between dogs and cats, since these comparisons further stratify the data.
Thank you for these comments. We are aware of the limitations regarding sample size and number of treated animals. We also agree that there will be influences at veterinarian (setting) level. However, we do not have the correct data to make any statements regarding this expected bias, as this was not within the scope of the study. Nevertheless, we did add a general comment on this with a reference to a recent study elaborating on the issue. Within the discussion, following adaptations were made:
“However, with the small sample size in this study, extrapolations of AMU and AMR to a national level should be done cautiously.” [lines 259-260].
“The biggest limitation of the collected data lies in the small sample size analyzed in our study, which makes extrapolation of the results to a national level challenging. In addition, the group of treated animals was small. Therefore, comparisons of AMU and AMR levels between countries and species should be interpreted while keeping in mind the restrictions of this small sample size. Nevertheless, the results did show similar trends with previous studies [25,26,28,45–48] and enabled the identification of some significant differences, although some might be missed due to a lack of power. A recent Dutch study reported on the many different factors influencing the antimicrobial prescribing behaviour of veterinarians [52]. As no inclusion criteria were present at the level of the veterinarian, it is expected that a certain (selection) bias is present within the study.” [lines 288 - 297]
Sequencing data would have been ideal, with some comparisons of resistance mechanisms and phylogeny with E. coli from other sources, but this is not an absolute requirement to meet the study’s goals.
We agree with the Reviewer that such data would have been of interest. However, the main focus within the EFFORT project were broilers and pigs, for which E. coli were sequenced and faecal samples pooled at farm level for metagenome analysis. Similar analysis was not performed for isolates and samples collected from companion animals.
Specific comments:
Abstract: Should add mention that no association between AMU and AMR was found, while cautioning that several factors complicate this analysis. Still, it is an important finding given that AMU and AMR are measured in this study, and should be mentioned in the abstract.
We added a brief sentence on this in the abstract. It now reads: “Most common resistance was to ampicillin (18%). No association between AMU and AMR was found in the investigated sample. The issue to address, regarding AMU…” [lines 37-38].
L96-8: Should more clearly define in text and/or table that treatment incidence is the % of the year that a dog or cat is on average treated. Recommend rephrasing to say something such as:
Cats were treated from 0.0-16.6% of the year’s duration, and 0.0-13.4% of the year for dogs. On average, dogs were treated for 0.9% of the year (3.3 days), and cats 0.6% (1.8 days), with these numbers reflecting the treatment incidence (TI) in Table 1.
We agree that the text did not emphasize the meaning of treatment incidence enough. We adjusted the manuscript according to the comment, with some minor adaptations. The revised manuscript now reads: “Taking into account the total sampled population (i.e. treated and not treated), treatment incidence (TI), which resembles the percentage of a full year that the animal has been treated with a standard dose of antimicrobials, ranged from 0.0% to 16.7% in cats and from 0.0% to 13.4% in dogs. The mean treatment incidence (TI) was 0.9% for dogs and 0.5% for cats, meaning that, on average, dogs and cats included in this study were being treated with a standard dose of antimicrobials during 3.3 days (0.9%) and 1.8 days (0.5%) in a full year, respectively.” [lines 97-100]
L100: This line belongs in the previous paragraph, where it was already stated that 25% of cats in Belgium were treated.
Thank you for noticing this. We should indeed avoid repetition, so we removed this sentence [lines 102-103.
L109-110: This sentence isn’t clear, as it makes it sound as if cefalexin and spiramycin-metronidazole were the ‘most chosen products’ instead of amoxi-clav
We understand the confusion and adjusted the sentence accordingly. The revised manuscript now reads: “In dogs, cefalexin and spiramycin-metronidazole (11% both) were the most chosen products, after amoxicillin-clavulanate.” [line 113]
Figure 1: Check each of the drug names; I think at least one is wrong (is it supposed to be trimethoprim-sulfamethoxazole?), with several having an extra ‘e’ at the end orbifloxacin, ofloxacin, clindamycin).
Thank you for noticing these errors, we have adjusted figure 1 accordingly [see figure 1].
L122: Would not recommend saying ‘longer’, considering these differences are not statistically significant. Would simply state the average durations without comment.
Thank you for this comment. We adjusted the text and the revised manuscript now reads: “Average treatment duration equalled 48 days for ofloxacin (one prescription), 19 days for metronidazole (2 prescriptions) and 14 days for cefovecin (7 prescriptions). The average treatment duration for oral administered antimicrobials equaled 10 days (avg ± sd = 10 ± 7 days; min-max = 1 – 42 days; 55 prescriptions), compared to 9 days for topical antimicrobials (avg ± sd = 9 ± 11 days; min-max = 1 – 42 days; 11 prescriptions) and 6 days for parenteral antimicrobials (avg ± sd = 6 ± 6 days; min-max = 1 – 14 days; 18 prescriptions) [lines 122-128]
L132: Don’t understand the point being made with this clause that begins ‘with significant differences…’. Recommend rephrasing.
Thank you for this comment. The significant differences refer to the OR and 95% CI mentioned after the sentence, regarding the odds to find a full susceptible isolate. Based on this comment, this was not clear. Therefore we adjusted the text and the revised manuscript now reads: “At country level, Italy had a higher number of resistant isolates (41%) than Belgium and the Netherlands (23% and 17%, respectively). As a result, when comparing to Italy, the odds to find a full susceptible isolate in Belgium and the Netherlands, were 2.7 (95% CI: 1.4– 5.1) and 3.9 (95% CI: 2.0 – 7.8) times higher, respectively (a complete overview is shown in Table S3). This shows significant differences in terms of full susceptibility between Italian isolates on one side and isolates from Belgium and the Netherlands on the other side [line 135-140]."
L149: This definition of MDR as three drugs is different than in the methods, which lists three or more classes. Please clarify.
We have changed the wording in M&Ms and in the Results so that now it is the same in both sections: “resistance to three or more antimicrobial classes/subclasses”. This is because the panel of antimicrobials (panel 1 vs Enterobacteriales as in the Dec 2013/652/EU) is specifically designed to contain drugs which are representative of different classes/subclasses, and resistance towards each drug class/subclass represents an indicator of different genetic basis of resistance.
This is true of course if you have used some basic expert rules in counting MDR patterns in the two critical cases: fluoroquinolones (CIP; NAL: R to CIP AND NAL counts 1) and beta-lactams: AMP; TAZ OR FOT: R to (FOT OR TAZ) AND R to AMP counts 1, since the genetic basis of 3rd 4th gen resistance inactivates also AMP. I hope I have clarified this point. [lines 156 & 357]
L196: add: ‘, respectively’ at end of this sentence to clarify to which animals the 9.0 and 9.2 refer
Thank you for this comment, we adjusted the text accordingly. The revised manuscript now reads: “Previous studies on European broiler and pig farms, using a similar quantification method, showed a median TI of 9.0 and 9.2 per 100 animal-days at risk, respectively [43,44].” [line 203]
L204: Unclear what ‘Next to common characteristics’ refers to; recommend deleting
We agree, this part of the sentence was deleted. The revised manuscript now reads: “AMU showed significant variation between countries…” [line 211].
Discussion: Need to mention overall limitation of AMU data, with only 20 cats and 38 dogs treated overall. This means some variation in the data and additional limitations when further stratified by country.
(see answer to first comment)
L232: E. coli with mcr-1 was also found in a dog in Germany: https://www.ncbi.nlm.nih.gov/pubmed/28122910
Thank you for this addition, we were not aware of the sampled dog faeces in this study. We added Germany to the list. The revised manuscript now reads: “To the best of our knowledge, colistin resistance in companion animals has only been described in China (2015) [9], Germany (2017) [56], Finland (2018) [57], Ecuador (2019) [58] and now also in the Netherlands.” [lines 239]
Add discussion of AMU in people, since they may be more likely to share microbes than agricultural animals with companion animals (especially since animals on farms were not included in this study).
The following sentence was added to the discussion: “Further research, focusing on differences at species and country level in AMU and AMR in companion animals will be essential to answer these questions. In addition, future scientific research should also look into the AMU and AMR levels of the owners, as different studies have already suggested associations regarding AMR between companion animals and their owners [18,60–63]. [lines 272-274] A more elaborate discussion does not seem relevant within the scope of this study.
L262-5: I do not understand this sentence, since the ESVAC reports focus on food animals, correct? It is therefore not expected for that report to be relevant to this study focusing on companion animals.
The ESVAC reports focus on food animal production, but also report on the use of tablets separately as an estimation of the usage of antimicrobials in companion animals. In both cases (food animal production and companion animals), Italy is a much higher user than the Netherlands and Belgium. Nevertheless, we find low usage in Italian cats together with high resistance levels. In this part of the discussion we want to suggest a possible link between our AMR results in Italian cats and the ESVAC numbers, being that a general high usage in veterinary medicine could be driving these AMR results. And in such a case, why is there a difference between cats and dogs? To clarify this part, we made the following adjustments to the text [lines 259-274]:
“More importantly, average TI in Italian cats (0.3) was the lowest compared to 0.5 in the Netherlands and 0.9 in Belgium. However, with the small sample size in this study, extrapolations of AMU and AMR to a national level should be done cautiously. In fact, general veterinary AMU levels in Italy do exceed those in Belgium and the Netherlands considering data from the ESVAC reports covering all veterinary antimicrobials sold at national level in 30 European countries [59]. These higher usage levels in Italy, as reported by ESVAC, are present for both the reporting on sales of tablets (used in companion animals) and the reporting on other pharmaceutical forms (used mainly in food-producing animals, including horses) [59]. This raises the question if the higher prevalence of AMR and MDR observed in Italian cats, together with lower odds to find a fully susceptible Italian isolate compared to Belgium and the Netherlands, could be explained by this general higher AMU in veterinary medicine in Italy, and in such a case, why Italian cats seem to have higher AMR and MDR levels compared to dogs. The latter is not the case for the Netherlands, where dogs show higher levels of AMR and MDR,…” .
And in lines [276-278]:
“The fact that AMU levels in this study do not completely comply with the trends seen in those numbers of the ESVAC reports representing the use in companion animals (sales of tablets [59] …”
L372: ‘levels of AMU in companion animals seem rather low’ is an ambiguous statement of limited value. I recommend stating that ‘x% of animals did not have antimicrobial usage over a one year period. However, when they were used, they were often CIA…’
We agree with this comment and added the recommendation to the text with some additional adaptations to make it clear for the reader and avoid misinterpretation. The revised manuscript now reads: “In conclusion, with 81% of the animals not receiving any antimicrobial treatment over a year, AMU in companion animals seems rather low compared to AMU on pig and poultry farms[43,44]. Nevertheless, when an antimicrobial was used, it was often a CIA” [lines 377-379]
Need to add the additional limitations of not including different types of veterinary care facilities (e.g. animal hospitals, etc) that may have dramatically different AMU (including with drugs administered IV, etc).
(see answer to first comment)